# State/Academia Key Stakeholders' Perceptions Regarding Bioeconomy: Evidence from Greece

**Marios Trigkas * and Glykeria Karagouni**

Laboratory of Forest Economics, Marketing, Innovation & Entrepreneurship, Department of Forestry, Wood Sciences & Design, University of Thessaly, 43100 Karditsa, Greece; karagg@uth.gr
* Correspondence: mtrigkas@uth.gr; Tel.: +30-24410-64721

**Abstract:** While researchers of the area try hard to conceptualize the bioeconomy, it appears that it is harder for the variety of stakeholders to agree on the critical elements that form sustainable bioeconomy models. The aim of the present paper is to gain insight into major players' understanding of the bioeconomy concept to form policies and strategies or direct education and research. Using data collected from academia and state stakeholders in Greece, this paper articulates perceptions regarding the concept of bioeconomy from an academic, technological, and economic point of view. According to the results, the bioeconomy concept seems to be related to applied life and economic sciences, and engineering and technology sciences. Its technological interpretation regards innovation, new product development, and technologies. Empirical findings indicate an almost catholic acceptance of bioeconomy as an economic activity, no matter the science field or the state position of the stakeholders. They also highlight a clear need for synergies and a coherent cross-sectoral and interdisciplinary approach to produce novel knowledge, skills, technology, and innovation. The research contributes to the existing debate on the buzzing concept of the bioeconomy and fills a scientific gap at the regional level of a typical Mediterranean economy, enriching the related literature.

**Keywords:** bioeconomy; bioeconomy concepts; bioeconomy perceptions; innovation; sustainable development

## 1. Introduction

According to the European Union's (EU) definition, "bioeconomy comprises those parts of the economy that use renewable biological resources from land and sea—such as crops, forests, fish, animals, and micro-organisms—to produce food, materials, and energy [1]. The goal is a more innovative and a low-emissions economy, reconciling demands for sustainable agriculture and fisheries, food security, and the sustainable use of renewable biological resources for industrial purposes, while ensuring biodiversity and environmental protection [1]. More specifically, the bioeconomy includes agriculture, forestry, fisheries, and food industries along with traditional industries dealing with biomass utilization such as the paper industry, as well as parts of the chemical, biotechnology, and energy industries [1,2]. Today, more than 40 countries have incorporated the bioeconomy concept into their political agenda [3], indicating a global effort to transit into a sustainable biobased economy [4].

Although the term bioeconomy was initially introduced by scientists dealing with the industrial impacts of the evolutions in biology, the main reason for which bioeconomy became a "mainstream" policy idea in Europe was the initiative to further promote and advance the concept of bioeconomy [2,5]. Therefore, it appears that the concept has received attention from more than one side. At a scientific and research level, bioeconomy lies on science and technology. According to many researchers, all official strategies envision a technology-based transition to the bioeconomy [6,7]. On the other hand, bioeconomy is considered a dynamic social transformation [8] and a revival opportunity for the

globe but especially for rural and isolated areas [9,10], introducing the political aspect of the transition.

According to Bran and Dobre [11], the cross-sectoral and interdisciplinary characteristics of the bioeconomy leads to corresponding approaches and initiatives aiming to address environmental and socioeconomic challenges. Furthermore, specific sciences such as biology, biotechnology, environmental and engineering sciences, economics, and information technology are mentioned as the main disciplines that formulate the concept of the bioeconomy [5]. In the same vein, new capabilities for the development of the bioeconomy are mirrored by the increasing demand and the rapid development for biobased products and markets, respectively [12,13]. New opportunities for business ventures, as well as the creation of supply chains and markets for biomass and biobased products, energy, and related services, are emerging globally highlighting the economic aspects and impact of the bioeconomy.

However, this transformation seems to be hindered by the resistance and the status of fossil-based industries in the global economy [4,6,14]. Hence, the transition to the bioeconomy model also constitutes a significant political issue, since it requires strong policy will and well-formed dynamic initiatives [15,16]. According to the relevant literature, the implementation of contemporary methods for supporting the development of skills and capabilities among policymakers constitutes a key element for the foundation of the bioeconomy pillars [17–19].

With regard to the economic aspects related to the bioeconomy models, there seems to exist an ongoing discussion on how to measure and determine their contribution, due to the cross-sectoral nature of the bioeconomy, the rather restricted way of establishing terms of evaluation [20], the inability to understand the full scale of potential direct and indirect benefits for a country [21], or even just because of the lack of credible statistical data [22]. These weaknesses constitute obstacles for the introduction of integrated policies, governance, and strategies at the regional, national, and international level [15,23].

From the above, it is quite evident that crafting a common strategy for the transition to the novel bioeconomy models is not easy at all. While researchers of the area try hard to conceptualize the bioeconomy, it appears that it is harder for the various types of stakeholders to agree on the critical elements that form viable and sustainable bioeconomy models [7]. Perceptions, interests, and beliefs seem to vary depending on the positioning of the stakeholders; for example: is it food or technology?

In accordance, policymaking and strategy building is heavily impacted by the perceptions of state and academia/research stakeholders; the former stakeholders involve the decision makers with the potential to implement laws, directives, and policies, while the latter stakeholders drive research and facilitate technology, as well as the innovation and, thus, production of biobased goods, in addition to constituting the main sources of education and training toward a more sustainable way of living. Both categories of stakeholders facilitate business and support citizen and consumer transition to the bioeconomy.

The aim of the present paper is to gain insight into major players' understanding of a bioeconomy and to explore how they approach the bioeconomy concept in order to form policies and strategies or direct education and research within the Greek context. For that purpose, academia and state stakeholders were selected as a sample to be asked about their perception regarding the concept of bioeconomy and the relevant models from an academic, technological, and economic point of view. We argue that the findings of the present research contribute to the existing literature of the bioeconomy, conceptualizing its paths as a model for shifting to a more sustainable and inclusive economy, especially in emerging and marginal aspects for the concept delimitation of environments such as the Greek state/academia context. The paper purports that further research on the issue can help related similar policy, economic, and social contexts internationally to develop more oriented bioeconomy strategies, in alignment with the already developed strategies at EU and international level.

Thus, this paper tries to give concrete answers to the following research questions:

Q1: What are the main perceptions of Greek key stakeholders in the state and academia/research categories regarding the very concept of bioeconomy?

Q2: How do Greek key stakeholders in the state and academia/research categories understand the relation of bioeconomy to technology?

Q3: How do Greek key stakeholders in the state and academia/research categories understand the bioeconomy as "an economic activity"?

We argue that our research contributes to the existing debate on the buzzing and emerging concept of bioeconomy internationally and fills a scientific gap at the regional level of a typical Mediterranean economy, enriching the related literature.

## 2. Materials and Methods

### 2.1. Data Collection

The primary research focuses on Greek bioeconomy key stakeholders, which are environmental, economic, and technological universities and research organizations, state-related entrepreneurship supporting organizations, and public authorities such as ministries and civil services dealing with natural environment, economy, and development.

More specifically, for the universities, we used the 2020 official State of Preference Form of the Greek Ministry of Education that the candidate students each year fill under the framework of the exams they undertake for higher education. We identified 152 university departments (from a total of approximately 500) in the fields of natural environment, economic studies (economics, entrepreneurship, business administration, marketing, etc.), and technological orientation (engineering, ICT, materials technology, etc.).

To identify the state- related entrepreneurship supporting organizations, we used the data from the Enterprise Europe Network—Hellas (EENH). This is a network of integrated entrepreneurial support, which is constituted by research and technology organizations, industrial associations, chambers of commerce and industry, governmental SME agencies, and acknowledged organizations in the sectors of innovation and small and medium-sized enterprises (SME's). EENH comprises 12 industry associations distributed all throughout Greece [24]. For the purposes of the study, we selected only state-related organizations such as the chambers, which are public law entities, and institutes that report to the general secretariats of relevant ministries [24,25].

Our survey also included the main public authorities dealing with the natural environment, economy, and development in Greece, namely, the Ministries of Financials, Development and Investment, Environment and Energy, Rural Development and Food, and Education and Religious Affairs. Lastly, we addressed the main research centers in Greece which are under the supervision of the General Secretariat of Research and Development (GSRT) of the Ministry of Development and Investments, i.e., 11 research centers [25]. Thus, our sample consisted of 175 organizations in the above categories of stakeholders, ensuring a broad understanding of the investigated issues included in the questionnaire.

For the purposes of the survey, we used as our primary research tool a structured questionnaire, specially designed to approach and derive the structural components of the complex and buzzing bioeconomy concept, according to the related literature and previous research on the issue. The data were collected using the Google Forms app. The survey period was between March and August 2020. A total of 78 questionnaires were collected, 71 of which were taken for evaluation, since seven of them were not sufficiently completed and were excluded from our analysis. This resulted in a response rate of 40.6%, which is considered acceptable, since the response rate does not assess the response bias or quality of research [26,27]. The type and main activity of the surveyed participants are presented in the following Table 1.

**Table 1.** Profile of the surveyed key stakeholders.

| Type of Key-Stakeholder | Percentage % | Main Activity | Percentage % |
| --- | --- | --- | --- |
| University—research center | 69.0 | R&D | 70.4 |
| | | Education and training | 54.9 |
| | | Manufacturing—services | 12.7 |
| | | Technology supplier | 4.2 |
| | | Technology services provider | 11.3 |
| Ministry—public authority | 9.9 | Public administration | 12.7 |
| State-related Business support organization | 7.0 | Professional business networking | 12.7 |
| | | Financial support | 4.2 |
| Other | 14.1 | Other | 8.5 |

*2.2. The Questionnaire and Method*

Our questionnaire items were developed using five-point Likert scales. The development of scales was based on a relevant literature review, as well as empirical and theoretical contributions from bioeconomy scholars in Greece [4,19,28,29]. The questionnaire was also sent to members of the Greek Bioeconomy Forum, a think tank on the concept of the bioeconomy in Greece, including several academics, scholars, and professionals in fields related to bioeconomy.

The final questionnaire consisted of 16 questions grouped into five main categories. The first group included three general questions about the level of the respondent's involvement with any type of bioeconomy; the second group explored the stakeholders' perspective of the concept within an academic context (two questions of 20 items); the third group questioned the technological understanding of bioeconomy (two questions of 23 items); the fourth group evaluated the economic aspects of bioeconomy with three questions of 43 items; the final group of questions regarded the profile of the surveyed organizations.

The data were recorded, processed, and analyzed using the IBM SPSS Statistics 25 special statistical program, as well as the relevant frequency checks (Frequencies), descriptive statistics, and correlations [30].

The questionnaire was tested foe its internal consistency to verify its reliability on the measurement of the different bioeconomy concepts. We used the Cronbach's alpha coefficient for the homogeneity of the scales [31]. Results of Cronbach's alpha $\geq 0.7$ are acceptable for the reliability of internal consistency [32,33].

To assure the content validity of the questionnaire, the required assessment was made before the collection of the data [34]. Thus, we initially determined the concept of the different bioeconomy approaches, and we detected the dimensions that constitute the determined variables, to include them in our questionnaire's items [35]. Afterward, the questionnaire was tested by a group of bioeconomy experts, including university professors and researchers—all members of the Greek Bioeconomy Forum—to evaluate the appropriateness of the sum and the content of the items included, which are directly correlated with the meaning that was under investigation, as well as the suitability of the scale content [36]. Furthermore, this questionnaire's pretesting led to changes, such as regrouping and reformulating some questions, to reduce the size of the questionnaire, since the majority of the respondents considered the difficulty and the time length to complete the questionnaire a little complex.

For the determination of the main dimensions (factors) that measure the approaches under investigation of the bioeconomy concept by the surveyed key-stakeholders, we used factor analysis to reduce the questionnaire's factors that describe these approaches. The correlation of the questionnaire's items was the criterion used.

Construct validity was also tested using factor analysis to seek the groups of the questionnaire items that were conceptually and statistically related to each other [37].

Using factor analysis, we tested if the items of the questionnaire statistically belonged to the dimensions, i.e., the factors used to describe the different bioeconomy concept approaches.

Eigenvalues were used to ensure that each item was distributed with a high load to each of the main factors [32,38]. Factor loadings also showed the correlation of each item with a main factor that emerged from the analysis. Items with high factor loadings (over 0.3) in the rotation component matrix were selected as the main ones that significantly contributed to the description and determination of the main factors that emerged to describe the concept of bioeconomy under different approaches.

Furthermore, the content reliability of the dimensions that emerged was tested using Cronbach's alpha coefficient. Dimensions with a high value of Cronbach's alpha (near or over 0.7) were considered reliable [32].

### 3. Results

*3.1. The Concept of Bioeconomy within an Academic/Research Context*

Our analysis revealed that four key factors of total variance accounted for each factor (Table 2):

(1) A tension to relate bioeconomy to certain scientific areas, more precisely, (1a) applied life and economic sciences and (1b) engineering and technology sciences.
(2) Emergent need for novel knowledge, skill development, and relevant policymaking.
(3) An association between bioeconomy and open innovation processes and models.

**Table 2.** Reliability analysis of academic context approach for the bioeconomy concept.

| No. | Factor | Overall Cronbach's Alpha | Determinants | Cronbach's Alpha If Item Deleted |
|---|---|---|---|---|
| 1 | (1a) Applied life and economic sciences | 0.850 | Focus on food science | 0.811 |
| | | | Focus on economic and social sciences | 0.816 |
| | | | Focus on ecology | 0.835 |
| | | | Focus on innovation and entrepreneurship | 0.820 |
| | | | Focus on agricultural science | 0.827 |
| | | | Focus on other life sciences | 0.853 |
| | | | Focus on bio/nanotechnology | 0.838 |
| 2 | Knowledge and skills development and relevant policy making | 0.867 | At level of institutionalization of economic strategy and policies | 0.848 |
| | | | At level of development in-house technical skills in businesses | 0.849 |
| | | | At level of turning applied sciences toward the creation of jobs with high specialties and skills in regard of bioeconomy | 0.853 |
| | | | At level of environmental policy | 0.863 |
| | | | At level of postgraduate/doctoral studies | 0.861 |
| | | | At level of pregraduate studies | 0.873 |
| | | | At level of households and individuals | 0.858 |
| | | | In R&D projects at businesses/industries | 0.852 |
| | | | In R&D projects at universities and research centers | 0.855 |
| | | | At level of industrial policy | 0.848 |
| | | | In implementing interdisciplinary education programs across the EU | 0.853 |

**Table 2.** *Cont.*

| No. | Factor | Overall Cronbach's Alpha | Determinants | Cronbach's Alpha If Item Deleted |
|---|---|---|---|---|
| 3 | Open innovation processes and models | 0.821 | In R&D projects at businesses/industries | 0.767 |
| | | | In R&D projects at universities and research centers | 0.786 |
| | | | At level of industrial policy | 0.764 |
| | | | In implementing interdisciplinary education programs across the EU | 0.788 |
| 1 | (1b) Engineering and technology sciences | 0.728 | Focus on engineering science | |
| | | | Focus on information and communications technology (ICT) science | |

The cumulative percentage of variance accounted for was 61.1%, meaning that the four factors (1a, 1b, 2, and 3) together accounted for 61.1% of the total variance. The overall Cronbach's alpha coefficient for all four emerged factors is presented in Table 2. The first factor (1a) reflected the focus of the bioeconomy concept to applied life and economic sciences (Table 2), with a Cronbach's alpha coefficient value 0.850, which is considered high enough; thus, our scale was reliable. Additionally, the Cronbach's alpha coefficients for all the determinants that our questionnaire included to shape this first factor were close to or lower than the overall Cronbach's alpha (Table 2—column "Cronbach's alpha if item deleted"); hence, we could include them in our analysis [32]. The results regarding all factors that emerged and described the academic context approach of the bioeconomy concept were similar.

### 3.2. Technological Approach

In the same vein, with regard to the ways that Greek key stakeholders of the state and academia/research categories understand the relation of bioeconomy to technology, the analysis initially resulted in six main factors with a total variance explained of 69.58%. However, in our results, we included four of them, since the overall Cronbach's alpha coefficients (Table 3) of two were significantly lower than 0.7. Thus, the factors were grouped as follows and the items constituting this grouping are presented in Table 3:

(1) New product development related to waste biomass and circular economy.
(2) High-added-value innovative products and energy.
(3) A focus on the promotion of the use of forest and agricultural biomass for products and energy.
(4) Networking and industrial symbiosis.

The scale that emerged was reliable according to the results of Table 3, since, for all four factors, the overall Cronbach's alpha coefficients were close to or greater than 0.7.

### 3.3. Perceptions of the Bioeconomy Concept as an Economic Activity

The statistical analysis initially indicated seven major factors with a total variance explained up to 72.1% However, according to reliability analysis, we ended with six main factors of the economic approach for the bioeconomy concept (Table 4) by the surveyed key stakeholders:

(1) Bioeconomy promotes synergies and economic development.
(2) Bioeconomy promotes innovation (as an economic activity).
(3) Bioeconomy relies on funding.
(4) Bioeconomy is related to strategic competitive advantage.
(5) Bioeconomy is based on economies of scale.
(6) Bioeconomy enhances competitiveness.

**Table 3.** Reliability analysis for technological approach of the bioeconomy concept.

| No. | Factor | Overall Cronbach's Alpha | Determinants | Cronbach's Alpha If Item Deleted |
|---|---|---|---|---|
| 1 | New product development in relation to waste biomass and circular economy | 0.901 | Utilization of urban waste biomass under the circular economy implementation context | 0.874 |
| | | | Utilization of waste forest and agricultural biomass under the circular economy implementation context | 0.878 |
| | | | Utilization of waste industrial biomass under the circular economy implementation context | 0.885 |
| | | | Use of renewable raw materials for the production of basic chemical construct elements | 0.880 |
| | | | Design and development of biodegradable products | 0.888 |
| | | | Utilization of aquaculture and fisheries biomass | 0.899 |
| | | | Production of new construction and building materials | 0.901 |
| 2 | High-added-value innovative products and energy | 0.803 | Utilization of biomass for the production of bioenergy (biofuels and electricity) | 0.764 |
| | | | Bioeconomy focuses in biotechnology techniques for the production of animal food, pharmaceuticals, chemicals, and fuels | 0.711 |
| | | | Bioeconomy focuses on the establishment of biorefineries | 0.712 |
| 3 | A focus on the promotion of the use of forest and agricultural biomass for products and energy | 0.726 | Utilization of forest biomass for the development of new products and energy | 0.700 |
| | | | Utilization of agricultural biomass for food safety and energy reasons | 0.635 |
| | | | A biobased product should be constituted as a whole by renewable biological resources | 0.654 |
| | | | Bioeconomy is based on nontechnological innovations | 0.669 |
| 4 | Networking and industrial symbiosis as basis of new value chains | 0.713 | Bioeconomy is based on nontechnological innovations related to synergies | 0.675 |
| | | | Establishment of synergies, networking, cooperation, and industrial symbiosis are necessary for the development of the bioeconomy | 0.636 |
| | | | Differentiation in business value chain is a fundamental factor for the development of the bioeconomy | 0.567 |

**Table 4.** Reliability analysis for economic approach of the bioeconomy concept.

| No. | Factor | Overall Cronbach's Alpha | Determinants | Cronbach's Alpha If Item Deleted |
|---|---|---|---|---|
| 1 | Bioeconomy promotes synergies and economic development | 0.906 | Development of win–win cooperation | 0.886 |
| | | | Development of win–win synergies among organizations—enterprises and institutions | 0.887 |
| | | | Adoption of new business models for bioeconomy | 0.893 |
| | | | Open/entrance in new markets | 0.896 |
| | | | Import of new skills and knowledge in markets | 0.897 |
| 2 | Bioeconomy promotes innovation | 0.885 | Import of new skills and knowledge in markets | 0.856 |
| | | | Boosting of innovative products and processes | 0.873 |
| | | | Production of added value innovative products that better address consumers' needs | 0.869 |
| | | | Creation of new knowledge and skills | 0.865 |
| | | | Support of sustainable consuming and development of novel consuming models | 0.873 |
| | | | Enhancement of research and development | 0.873 |
| | | | Enhancement of innovation culture in businesses/organizations | 0.885 |
| | | | Development of new or supporting of existing innovation systems at micro/meso/macro level | 0.879 |
| | | | Boosting of innovative products and processes | 0.900 |
| 3 | Bioeconomy relies on funding | 0.847 | Securing funding with better utilization of funding tools and programs | 0.804 |
| | | | Leverage of private capitals for investments | 0.822 |
| 4 | Bioeconomy is related to strategic competitive advantages | 0.822 | Supporting of cascading use of resources | 0.783 |
| | | | Adoption of open innovation models by the enterprises | 0.771 |
| | | | Development of core capabilities for businesses/organizations | 0.780 |
| | | | Creation of new value chains for businesses | 0.805 |
| | | | Improving transfer of know-how for bioeconomy | 0.794 |

**Table 4.** *Cont.*

| No. | Factor | Overall Cronbach's Alpha | Determinants | Cronbach's Alpha If Item Deleted |
|---|---|---|---|---|
| 5 | Bioeconomy is based on economies of scale | 0.695 | Facilitating transfer of know-how for bioeconomy | 0.641 |
| | | | Creation of economies of scale in different industrial sectors | 0.552 |
| | | | Reduction in production costs | 0.612 |
| 6 | Bioeconomy enhances competitiveness | 0.722 | Better utilization of business/organization's resources | 0.689 |
| | | | Improvement of raw materials' supply chains | 0.830 |
| | | | Boosting employment and creation of new jobs | 0.807 |
| | | | Improvement of competitiveness through differentiation of organizations | 0.814 |
| | | | Improvement of products/services' quality | 0.582 |
| | | | Increase in market share for the enterprise | 0.618 |

*3.4. Main Barriers*

Lastly, we investigated the main barriers for the development of the bioeconomy concept and model in Greece. The evaluation of the given answers reveals that the most important barriers were as follows:

- The lack of a national strategy for bioeconomy ($4.23 \pm 0.89$).
- The resistance to change and the lack of innovation culture by the stakeholders ($4.20 \pm 0.90$) and
- The lack of awareness regarding the concept and the opportunities that bioeconomy offers to both businesses and citizens ($4.20 \pm 0.95$).

All the evaluated factors included to our results were evaluated at the scale of moderate to strongly significant.

Correlation analysis on the above barriers for the bioeconomy in Greece revealed several correlations among factors. The most significant ones are highlighted below.

- The lack of awareness campaigns for businesses/citizens and the lack of national strategy for bioeconomy ($0.383$, $p < 0.001$).
- The lack of an international/interstate governance system for the bioeconomy and the gaps and complexity of national legislation ($0.393$, $p < 0.001$).
- Resistance to change and lack of innovation culture by the stakeholders and the difficulties for cooperation among different economic sectors ($0.447$, $p < 0.001$)
- The high technological level of bioeconomy and the lack of know-how and untrained and skilled research and labor personnel ($0.543$, $p < 0.001$).
- The high technological level of bioeconomy, the lack of know-how, and the abandonment of rural areas and of the primary production ($0.439$, $p < 0.001$).
- The high technological level of bioeconomy and the lack of relative motivations for investing in bioeconomy ($0.398$, $p < 0.001$).
- The abandonment of rural areas and of the primary production and the lack of relative motivations for investing in bioeconomy ($0.479$, $p < 0.001$).
- The lack of relative motivations for investing in bioeconomy and the lack of national strategy for bioeconomy ($0.405$, $p < 0.001$).

## 4. Discussion

The development of bioeconomy plays a fundamental role for the transition of the European economy toward a green and sustainable economy, while it opens new potentials for entrepreneurship, research, and innovation. New and innovative entrepreneurial ventures for securing food, products, and energy, based on a rational utilization of biological resources, are entering in an implementation orbit across Europe and the rest of the world, opening new pathways for a climate-neutral and competitive economy. However, while the whole of Europe promotes and enhances the bioeconomy concept, a quite significant part of the stakeholders in several countries, such as Greece, do not seem to be quite familiar with the bioeconomy meaning and its entrepreneurial potential yet [29]. Thus, the concept of the bioeconomy, since it still stands as an emerging meaning for most countries, including Greece too, has become a field of different and sometimes contradictorily perceptions and understandings by the different categories of stakeholders. This depends on the specific characteristics that each category is incorporating; therefore, bioeconomy still constitutes an interdisciplinary and cross-sectoral concept with its complexity rather magnified [5,39,40]. Hence, it appears that there are many different interpretations of the very concept of bioeconomy, depending on the category of stakeholders. Scientists relate it to research and specific fields while economists refer to novel perspectives on economics; on the other hand, bioeconomy has further become a buzzword by state members, public organizations, and institutions to promote a supposed socioeconomic and ecological transition.

Regarding the first research question of the study (Q1), our findings (Table 2) are quite consistent with the core line of the relevant literature with regard to the different ways stakeholders tend to understand and frame bioeconomy [5,8,11,39,41].

More specifically, according to the answers retrieved (Table 2), the bioeconomy concept in an academic and policy context seems to be related to specific scientific areas and more precisely to applied life and economic sciences and engineering and technology sciences. This is quite normal since bioproducts derive from research in these fields; however, it is imperative to break silos among scientific fields to succeed in synergy creation, novel bioeconomy value chains, and networking which appear to be fundamental when viewing bioeconomy from a technological and economic perspective (Tables 3 and 4). Further integrating the findings (Table 2), it appears that higher education of environmental, technology, and economic studies should diffuse among all academic disciplines, focusing on adjusted knowledge and skill development, along with the introduction of novel innovation models and networking strategies considering individual, consumer, and organization perspectives.

The same indications also regard research collaborations. The results (Table 2) indicate a clear need for synergies among entities working on different research issues, i.e., a need to share, knowledge, skills, and information, as well as collaborate. Yet, these critical fields of public policy continue to operate in silos, since key stakeholders have different perceptions on critical issues considering the essence and potential of bioeconomy. Thus, an emphasis on and support of research that takes place in the higher-education and research ecosystem of Greece, in combination with private research initiatives, could shape the required space for innovative solutions under the bioeconomy concept.

The findings (Table 3) clearly demonstrate that novel knowledge, skills, technology, and innovation can be achieved under the development of synergies among different scientific and industrial fields and networking of stakeholders, a fact that reflects the importance of open innovation models and processes for the bioeconomy [42–45].

With regard to the second question, the findings (Table 3) indicate that the technological interpretation of the bioeconomy concept by the Greek key stakeholders is related to innovation and the development of new products and technologies (usually under the circular economy scheme, focusing mainly on the utilization of biomass for energy and launch of new biobased products). The results are also in line with the academic context approach [4,44–46]. Furthermore, it appears that Greek academia and state stakeholders prioritize forest and agricultural biomass (Table 3), which is quite normal, since the two

sectors, together with the blue economy, constitute the main ones in Greece [29]. In the same vein, the concept of industrial symbiosis has seemed to gain ground within Greek industrial ecosystems; a significant number of respondents of different areas consider it a major contributor to the implementation of bioeconomy models in Greece (Table 3).

Thus, regarding the Greek economic reality, our results (Table 4) indicate that bioeconomy is welcomed as the next wave for economic growth, and it is anticipated to offer significant opportunities for job creation, opening of new markets, and improvement of competitiveness. According to the findings (Table 4), there is an almost catholic acceptance of bioeconomy as an economic activity by the respondents, no matter the science field or the state position they own. The majority sees the importance of the bioeconomy role in the socioeconomic development of Greece. Interestingly, they also relate networking and synergies to economic growth; these two factors, together with innovation, also emerged as very important in the other two research questions (as referred above). In the same vein, but less strongly accepted, economies of scale and funding schemes seem to be considered as important elements when practicing a shift from traditional to bioeconomy models and when forming the new strategic competitive advantages of organizations (Table 4). This finding was quite expected by academia stakeholders of certain fields (e.g., engineering) However, it was not expected by respondents of economic sciences and all state stakeholders; therefore, a deeper analysis and further investigation is needed, perhaps further engaging the side of business. The study also revealed a gap among the investigated sides and the business category which is also supported by the finding that most stakeholder categories operate in silos, eliminating the opportunities to converge around perspectives and desired outcomes.

Overall, the findings indicate the focus by all state/academia stakeholders on overcoming traditional entrepreneurial and economic culture and a correlation of the bioeconomy concept to open systems with regard to knowledge, innovation, research, and synergies that are capable of improving competitiveness of businesses and economies.

In parallel, our findings (Table 3) ascertain the integration of bioeconomy and circular economy within the greater context of the green economy, in line with the current literature [28,40,46,47]. Greek bioeconomy is strongly connected to biomass utilization and the sustainable and efficient use of natural resources. Enhancement of information, knowledge, and skill development efforts constitutes a critical element of policymaking in combination with technological and production evolutions. According to this study, the Greek version of the implementation of the bioeconomy (and its extended meaning as a circular bioeconomy) could be achieved through the breaking down of silos, the pluralistic cooperation among different scientific fields, technologies, institutions, and organizations, and direction for synergistic approaches. Such initiatives should be supported by relevant funding schemes, since the creation of new value chains are in need of significant investments, research support, development of innovations, and new circular economy pathways [42–44].

Hence, we argue that bioeconomy remains a complex meaning with a multidisciplinary background for knowledge and skill creation at different academic and social levels, relying on open innovation processes, which is also supported by the related literature [42,45,48]. However, it can undoubtably lead to social welfare through the establishment of new and differentiated values chains for biological renewable resources in products and energy production, addressing major environmental challenges and promoting sustainable entrepreneurship and economic growth.

The research findings (Tables 2–4) demonstrate an imperative need for cross-sectional, cross-discipline teams, in order to form a national bioeconomy strategy; specific policies could refer to better and more interdisciplinary information on the concept, its opportunities, and types of required transformations at the entrepreneurial, technological, legislative, and development level. The determinants of the bioeconomy concept and the bioeconomy barriers detected allow for the following recommendations for the development of a national bioeconomy strategy in Greece:

(a) Coherent policy commitment.

(b) Effective line-up with regional, national, and European policies.

(c) Investing in the acquisition and creation of relative new knowledge and innovation at all levels of the cross-cutting nature of bioeconomy, breaking down silos, and forming heterogenous teams for decision and policymaking in an effort to cover the extent of different ways bioeconomy is understood and framed.

This is a way to eliminate inconsistencies among science fields, research priorities, policies, initiatives, and legislation. A coherent cross-sectoral and interdisciplinary approach of knowledge, skills, and capabilities can facilitate the integrated development of the bioeconomy under criteria for environmental, social, and economic attribution, with the creation, utilization, and understanding of new technologies and local individualities [49–52]. In this context, research and innovation projects, organized education, training, and establishment of mechanisms for know-how transfer could constitute valuable tools for structuring a suitable environment for the bioeconomy development. Crafting strategies in such a way can allow for a conscious shift of production and economic activity from "sectors" to "systems" or "value chains", through the creation of constructive networking paths for the direct economic and production sectors within the context of biological resources [53].

## 5. Conclusions

The transition from a fossil-based economy toward a more biobased and sustainable economy constitutes one of the prevailing solutions globally for the confrontation of the urgent environmental, economic, and social challenges. The cross-cutting nature of the emerging biobased economy models causes a variety of perceptions and interpretations of the bioeconomy concept; these depend on the scientific field, the environmental and technological dimensions, and the socioeconomic impacts that this emerging model incorporates. On the other hand, the perceptions of the key stakeholder groups regarding the bioeconomy have a significant impact on the creation of appropriate frameworks regarding the development and implementation of the relevant transition models.

The research contributes to the existing debate on the buzzing concept of the bioeconomy and fills a scientific gap at the regional level of a typical Mediterranean economy, enriching the related literature. The empirical findings indicate the need for a coherent cross-sectoral and interdisciplinary approach to the knowledge, skills, and competences for the proper development of the bioeconomy in terms of social, economic, and environmental performance, creation, exploitation, and understanding of new technologies, as well as local specificities. In this context, research and innovation programs, organized training, and know-how transfer mechanisms can be valuable tools for building the proper framework for the transition.

**Author Contributions:** Conceptualization, M.T. and G.K.; methodology, M.T.; software, M.T.; validation, M.T.; formal analysis, M.T.; investigation, G.K.; resources, M.T.; data curation, M.T.; writing—original draft preparation, M.T. and G.K.; writing—review and editing, G.K.; visualization, M.T.; supervision, M.T.; project administration, M.T.; funding acquisition, no external funding. All authors have read and agreed to the published version of the manuscript.

**Funding:** This research received no external funding.

**Informed Consent Statement:** Informed consent was obtained from all subjects involved in the study.

**Conflicts of Interest:** The authors declare no conflict of interest.

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
