# Peer review of "State/Academia Key Stakeholders’ Perceptions Regarding Bioeconomy: Evidence from Greece"

_sustainability, doi:10.3390/su15139976_

Round 1

Reviewer 1 Report

This is an interesting study that the authors provide insight into major players’ understanding of a bioeconomy and to explore how they approach the bioeconomy concept to form policies and strategies.

The paper structure is appropriate. The introduction and research method sections were well written. The authors clearly explained the significance of the research topic, the gaps, and the importance of outcomes. The results were well presented and discussed with the results of other studies.

Reviewer 2 Report

It is necesary to have improvements according to comments in the text.

Reviewer 3 Report

It is my pleasure to review the manuscript (sustainability-2404199) submitted to Sustainability by Marios and Glykeria. In this article, the authors studied a questionnaire-based public opinion/survey of Greece for State/ Academia key-stakeholders’ regarding the bioeconomy. The major aim of the study was to fill a scientific gap at the regional level of a typical Mediterranean economy, enriching related literature. However, the manuscript is unnecessarily lengthy in several sections. Here are some observations to improve the article further: 

[1] Abstract: The authors have mentioned the background/rationale of the study, whereas methods, results, and the significance of the current results are missing.

[2] Introduction: This section is very lengthy. Please reduce it to half (readers will lose attention with unnecessary lengthy statements/writings).

[3] Materials and Methods:

In Table 1, write 69.1 instead of 69,1  and so on….

[4] Results:

Need to check lines 309-315 (points 2 and 3 are missing; it is Table, not table).

[5] Discussion:

Please corelate/mention Tables in this section for better justification of discussion. For example, in line 404, please insert Table(s) number after ‘…….. our findings (Table…)…., etc.

[6] Conclusions: This section is unnecessarily lengthy. It must be shortened to a single paragraph (or a maximum of two paragraphs) indicating the significant results of the study and future prospects.

Some typos and grammatical errors must be checked and improved. Overall, the manuscript seems to be more descriptive and should concentrate on a questionnaire-based public opinion/survey of Greece for State/ Academia key-stakeholders’ regarding the bioeconomy.

Moderate editing of English language

Round 2

Reviewer 3 Report

The authors have improved/modified the manuscript (sustainability-2404199) nicely as per my comments. Some typos must be corrected during the next steps (in line 201, correct - Cronbach’s alphalpha coefficient). In the discussion part, ‘The results (Table 2), indicate’ should be ‘‘The results (Table 2) indicate’. Similarly, please check out the full discussion part.

Minor editing of English language required

Author Response

Response to Reviewer 3 Comments

Point 1: The authors have improved/modified the manuscript (sustainability-2404199) nicely as per my comments. Some typos must be corrected during the next steps (in line 201, correct - Cronbach’s alphalpha coefficient). In the discussion part, ‘The results (Table 2), indicate’ should be ‘‘The results (Table 2) indicate’. Similarly, please check out the full discussion part.

Response: We would like to thank the reviewer for his final comments on our work, as well as the very detailed reviewing of our manuscript, significantly improving its quality. All relevant changes were made according to his final comments, after 2nd revision.